# Monitoring of Ion Mobility in the Cement Matrix to Establish Sensitivity to the ASR Caused by External Sources

**DOI:** 10.3390/ma15144730

**Published:** 2022-07-06

**Authors:** Michal Marko, Petr Hrubý, Martin Janča, Jakub Kříkala, Jan Hajzler, František Šoukal, Jan Vojtíšek, Martin Doležal

**Affiliations:** Faculty of Chemistry, Brno University of Technology, Purkyňova 118, 61200 Brno, Czech Republic; xchrubyp@fch.vut.cz (P.H.); xcjancam@fch.vut.cz (M.J.); xckrikala@fch.vut.cz (J.K.); xchajzlerj@fch.vut.cz (J.H.); soukal@fch.vut.cz (F.Š.); xcvojtisek@fch.vut.cz (J.V.); martin.dolezal7@vut.cz (M.D.)

**Keywords:** alkali–silicate reaction, diffusion, ionic mobility, cement, degradation

## Abstract

The possibility of the formation of an alkali–silicate reaction (ASR) is a crucial issue for the service life of concrete. The coexistence of key parameters such as the presence of alkalis, reactive SiO_2_, humidity, and temperature predetermine the possibility of its formation and application. When an ASR gel forms, it results in the concreting cracking and spalling as well as in the deterioration of its overall properties. The risk of ASR depends on the concentration of alkalis and their mobility, which influence their ability to penetrate the concrete. The objective of this study was to determine the ionic mobility of not only Na^+^ and K^+^, but Ca^2+^ as well, from external sources (0.5 and 1.0 mol/L solutions of Na/K carbonate, nitrate, and hydroxide) to a cementitious matrix as the precursor for ASR. The concentrations of ions in both the immersion solutions (ICP) and the cementitious matrix itself (SEM-EDX) were studied as a function of time, from 0 to 120 days, for leaching, and according to temperature (25 and 40 °C). The reaction products were characterized using SEM-EDX. Different diffusion rates and behavior were observed depending on the anion type of the external alkali source. Both sodium and potassium ions in all the three environments studied, namely carbonate, hydroxide, and nitrate, penetrated into the composite and further into its structure by different mechanisms. The action of hydroxides, in particular, transformed the original hydration products into calcium-silicate-hydrate (CASH) or ASR gel, while nitrates crystallized in pores and did not cause any changes in the hydration product. The driving force was the increased temperature of the experiment as well as the increased concentration of the solution to which the test specimen was exposed.

## 1. Introduction

Three conditions (i.e., reactive aggregates, moisture, and alkali) are necessary for the possibility of an alkali–silicate reaction (ASR) in concrete. By removing one of these, the possibility of ASR degradation has been nullified. However, there is almost no practical method by which to protect concrete from moisture access, and moisture affects the most valuable quality of concrete [1]. To date, the most effective method to limit ASR reactions has been by using non-reactive aggregates, but this is not always possible. Some regions can only access reactive aggregates for concrete production, and stone transportation over long distances is both economically and environmentally unsound [2]. Monitoring the alkali content in the materials during cement production is the most commonly used approach.

Clinker contains up to 2 wt.% (expressed in oxides) of alkalis [3,4,5]. On average, the alkali content is around 1 wt.%, originating from raw clay materials and feldspar. Cement clinkers in Europe typically have a higher potassium content by approximately 0.4–1.8 wt.% K_2_O more than sodium content, which is 0.05–0.5 wt.% Na_2_O. Clinkers containing K_2_O at the upper limit normally contain less-to-moderate amounts of Na_2_O [3,4,6]. However, the K_2_O/Na_2_O ratio varies over a relatively wide range from less than 0.25 to 10.0. Cement clinker also contains up to 2 wt.% of sulfates commonly calculated as SO_3_ [4,7]. During clinker firing, sulfates most often combine with alkalis [8].

However, there is generally an excess of alkali in the clinker. Excess alkalis, which are not bound in the form of sulfates, are present in the sintering zone in the melt of calcium aluminoferrite and the solid phases of alite and belite [9]. When the melt is cooled and crystallized, the alkalis preferably enter C_3_A (tricalcium aluminate), where depending on their concentrations, they affect the resulting crystal structure. Potassium, which is commonly found in clinker in higher concentrations than sodium, has a greater effect on the properties of clinker and, thus, on cement. It also passes into the C_3_A phase but always in combination with silicon or a trivalent iron atom. Similar to sodium ions, a change in the crystalline structure to orthorhombic was observed [10].

Materials based on a cement matrix are porous with pores from macro to nano in size. The content and size of the pores are dependent on the processing method and composition of the cement-based material. The pores are mostly the residue from the mixing water, where only part of it has been used to hydrate the clinker phases; the rest has been added to improve processability and will be subsequently evaporated [11]. In addition, the formation of pores in cement composites may also contribute to, for example, the formation of the hydration products in the calcium-silicate-hydrate (CSH) gel and their spatial characteristics [12]. The pores in the material are similar to interconnected capillaries. Through these capillaries, external substances can penetrate and react with concrete components such as aggregates and reinforcements. In addition to capillary pores, there may also be air pores that are larger in size [13].

The presence of pores increases the risk of penetration of undesirable species (in our study, these were represented by the various alkali sources) into the structure. These can have a negative impact on the overall properties and cause degradation. This penetration of external substances into the internal structure is driven by the various transportation processes [14]. For example, diffusion is associated with the concentration gradient, which forces the whole system to rebalance by migrating the external substance from a higher concentration to a lower concentration environment. The flow is governed by Fick’s laws of diffusion. Three other transport mechanisms coexist with diffusion. Permeability is a pressure-driven process where a fluid is forced into the concrete by means of increased external pressure, as compared to the lower internal pressure of the capillary system of the concrete. The increased pressure could be, for example, around the cementitious material when placed under a layer of water [15,16]. Capillary suction occurs due to the interaction between the fluid molecules and the capillary wall (e.g., the cementitious material). The interaction of these two environments is greater than the interaction between the liquid molecules themselves. Capillarity is not affected by ambient pressure. Finally, migration is a mode of transport where cations penetrating during diffusion must be balanced by corresponding opposing anions [17]. Otherwise, the whole element will become electrically charged, creating two oppositely charged poles. The electric field may be caused by different migration rates (mobility). The ions contained in the surrounding solution (pore) then migrate to the respective poles according to their charge. In a study concerning the migration of chloride ions, the Nernst–Planck equation, which complements Fick’s equations with the influence of the concentration gradient and electric forces, was used to describe the process. [18] These transport mechanisms can coexist and influence each other during the penetration phase and follow the mobility of a species inside the internal structure of the matrix. 

Current research in the field of ASR has been based on the consequences of this phenomenon itself, such as the influence of mechanical–physical properties and the persistence of composites and micromechanical laminary changes, as described in [19,20]. Based on the ASR mechanism [21], when there has been a gradual clotting of the formed gel and an increase in the interlaminar stress in the cementitious composite, there can be (and in practice, there is) a gradual formation of microscopic defects between the CSH gel structures and the interphases in the aggregate environment. As the ASR development increases, the microscopic defects then coalesce into macro-defects and cracks [22]. The external environment may then further enter these and accelerate the degradation of the composite by corrosion or erosion processes [23].

In the last two decades, research into the construction and technology of cement pastes, mortars, and concretes has considered the possibilities of reducing the risk of ASR. Individual risks (e.g., water coefficient, alkali deposition, cement type, aggregate reactivity) have been characterized, and methods to mitigate the risk have been outlined [24]. Aggregate reactivity has been assessed by the recognized standard ASTM C1293-08B [25]. Further research [26] has shown that supplementary cementitious materials, such as blast furnace slag, high-temperature fly ash, or other pozzolans, have reduced the risk of ASR by reacting with the portlandite and reducing the overall alkalinity in the long term. Traditional cementitious composites and alkali-activated materials based on pozzolans or latent hydraulic substances have been subject to ASR formation, as well as the traditional cementitious composites, due to the type of activator used where the most aggressive behavior of hydroxide activators with higher concentrations have been observed [27,28].

Few researchers have examined the processes associated with the exchange of alkali ions and their permeation of the cement composite structure. The research team in [29] analyzed the diffusion of lithium compounds (e.g., carbonate and nitrate), and [30] studied electrochemical migration and the replacement of sodium and potassium ions by deposited lithium ions. The migration of sodium and potassium ions through the body of the cement composite was described in the work of Xu and Hooton [31]. They, however, were also concerned with this process in an electric field. However, their conclusions showed a fundamental difference in the permeability of a given ion through the medium, which depended on the size of the ion and the type of anion of the medium. The sodium ion migrated through the medium faster than the potassium ion, and the carbonate medium slowed the migration, as compared to hydroxide. The influence of alkalis on the microstructure of cement composites and the distribution of super-fused alkalis inside the sample were investigated in [32]. The researchers observed that higher alkali concentrations were found when the sample surface had close contact with the environment. The alkali concentrations towards the sample’s body gradually decreased and showed a higher diffusion rate of sodium than of potassium ions.

The migration of sodium and potassium ions from the environment into cement composites has not yet been fully explored, particularly while monitoring changes in the alkali concentrations of both the solution and the sample, the changes in the microstructure, and the formation of secondary products. However, understanding how and to what extent alkali ions migrate as fundamental parameters affecting the vast majority of chemical processes is crucial to discovering the underlying causes of ASR formation.

The main objective of this study is to monitor the changes in the alkaline solution concentrations to which the cementitious composite test samples are exposed, to monitor the change in the distribution of alkali in the samples and to evaluate the changes on the surfaces of the test samples, as well as the changes in the microstructure and the possible formation of metamorphosed alkaline gels.

## 2. Materials and Methods

### 2.1. Preparation of Test Specimens and Their Characterization

Ordinary Portland cement of CEM I type 42.5R RC–road cement was chosen as the binding material. This type of cement contains 95–100% clinker and a maximum of 5% additional ingredients. At the same time, it is fast-setting cement, which is characterized by a higher increase in strength in the initial stages of hydration. It was produced by a Czech cement plant, Mokrá (Heidelbergcement group, Mokrá, Czech Republic). Its basic parameters were described as bulk density 3.130 g/cm^3^ and specific surface area, according to the Blaine method, 309 m^2^/kg. The characteristic chemical composition determined in accordance with the EN 196-2 is summarized in Table 1. The phase composition was determined by XRD (Empyrean PANalytical, Malvern Panalytical Ltd., Malvern, UK) with the following parameters for analysis: Cu Kα1 radiation (1.54059 Å), tube current 30 mA and voltage 40 kV, scan axis gonio, step size 0.01313° 2Θ, time per step 96 s, and scan range 4.5–90° 2Θ; the results were evaluated using the software HighScore Plus and are summarized in Table 2.

Cement paste with a water-to-cement coefficient (*w*/*c*) of 0.3 was used for the preparation of the 20 × 20 × 100 mm test specimens. This water coefficient corresponded to the normal consistency described in the standard EN 197-3. The basic characteristics of the hydration process were recorded using a calorimeter TAM Air (TA Instruments, New Castle, DE, USA). The measurement was based on standard ASTM C 1679. The measurement was conducted at the laboratory temperature −25 °C. The parameters for the determination of the setting process were determined using a Vicat needle and in accordance with EN 196-3. The prepared cement paste test specimens were demolded after 24 h of mixing and cured for 28 days in an environment with 100% humidity and a temperature of 25 °C. These cured specimens were subsequently used in further experiments. Measurement of the total porosity of the prepared mixtures and the porosity progression, including the dependence of the pore size on the calculated total porosity, was determined with a mercury intrusion porosimeter (PoreMaster 33, Quantachrome Instruments, Anton Paar, Macquarie Park, NSW, Australia) on the pre-prepared cement paste blocks with dimensions 7 × 7 × 21 mm (±0.1 mm). After 28 days, samples of hydrated and fully dried cement pastes (24 h of drying at 45 °C; then 24 h hours under laboratory conditions (25 °C at 40% humidity)) were also subjected to a water absorption test by immersion in a container of water. The weight changes associated with water absorption into the pores and structure of the material were recorded at time intervals of 10, 30, 60, 120, 180, 360, and 1440 min.

### 2.2. Alkali Solution Deposition

After the completion of the hydration process, the prepared samples were subsequently subjected to deposition into alkali solutions. Sodium and potassium alkali concentrations of 0.5 and 1.0 mol/L were used to study the ion exchange between the solution and the test sample. Na_2_CO_3_, K_2_CO_3_ (Penta Chemicals, Prague, Czech Republic, purity p.a.), NaNO_3_, KNO_3_, NaOH, and KOH (Lach-Ner, Ltd., Neratovice, Czech Republic, purity p.a.) as well as demineralized water (FCH BUT, CZ) were used to prepare these solutions. The sample was immersed in the specific alkali solution using polypropylene sealable containers. The ratio of sample weight to solution volume was 1:5. At selected time intervals, 1, 7, 28, 56, 84, and 120 days, changes in the concentration of the respective medium solutions were monitored using optical emission spectrometry inductively occupied plasma (ICP-OES) (Horiba Scientific–Ultima 2, Tokyo, Japan). In parallel, changes in the morphology of cement pastes and alkali distribution in these samples were characterized using scanning electron microscopy supplemented by EDX mapping (Zeiss EVO LS 10, DE, Oberkochen, Germany). The specimens were placed onto carbon tape, and the exposed fractured surfaces were sputter-coated with gold. The working distance in the measurement process was set to 12 mm, and the accelerate voltage was 15 kV. Moreover, the changes in porosity using the MIP were evaluated by the Poremaster instrument (Quantachrome Instruments, Boynton Beach, FL, USA). The working pressure range was from 0.14 to 231 MPa, which covered a pore diameter ranging from 6.5 to 1000 nm. These measurements were conducted using Hg with the surface tension of 0.480 N/m and a contact angle of 140°.

For clarity of the workflow, a schematic of the experiment is shown in Figure 1.

## 3. Results and Discussion

### 3.1. Characterization of Cement Paste Samples

The hydration process of the cement paste for road cement with a water coefficient of 0.3 is shown in Figure 2; the left graph shows the heat flow dependent on hydration time, and on the right, total hydration heat. The hydration peak occurs at 10 h and 39 min with a heat flux maximum of 3.4753 mW/g. The value of the heat of hydration of the cement used corresponded to the findings of [33]. The initial setting was 267 min, and the final setting was 341 min after mixing based on a Vicat needle test, according to the EN 196-3; 2016.

When a water absorption test (Figure 3 left) was performed on the road cement trap, the total water absorption (after 24 h of immersion) was 37.05 g of absorbed water. This value corresponded to a water absorption rate of 6.7%, relative to the weight of the body. The reason for the water absorption of the bodies was the capillary phenomenon in the pores, where pores of larger sizes were able to hold more water [34]. Moreover, the absorptivity of the body was one of the key aspects of the migration of alkali ions from and into the sample structure. 

The pore size distributions present in the prepared samples were plotted by the curve shown in Figure 3 right. For an overall evaluation of the results, only the regions between 10 nm and 5 µm could be used. Pore sizes above this interval were outside the calibration range of our instruments and may also continue to form clusters in the material leading to cavities or cracks. In general, they are not expected to occur in a compacted cementitious composite with good workability. However, pores up to 10 nm in size were the limits of the cavities where mercury could penetrate under pressure. The characteristic exponential shape of the calculated values of the total porosity of the prepared sample was similar to that of common cement pastes [35,36].

### 3.2. Monitoring the Migration of Alkaline Ions between Solution and Specimen

The behavior of Ca^2+^, Na^+^, and K^+^ alkalis was monitored in carbonate, nitrate or sodium, or potassium hydroxide solutions of a given concentration at 25 and 40 °C and using the ICP method. 

Changes in the concentrations of the alkali (sodium or potassium, and simultaneously, the concentration of calcium ions) were monitored so that both the processes associated with the exchange or substitution of ions in the structure of the samples monitored and the processes leading to the transformation of the CSH gel and the corresponding hydration products to CASH—C(Na)SH or C(K)SH could be characterized.

Only changes in the chemical composition of the alkaline solutions could be observed by ICP-OES, and a decrease in the alkali concentration was not necessarily related to diffusion into the structure or pores of the sample. It could also indicate various forms of crystalline and amorphous or gel-like products on the surfaces of the samples under investigation. Therefore, changes in the structure were also evaluated by SEM and EDX mapping. 

#### 3.2.1. Monitoring the Concentration of Calcium Ions 

According to the graphs of the behavior of calcium in solutions (Figure 4), two trends were observed. In the first, both the increases and decreases in concentration were more pronounced. The groups with more pronounced trends had a common denominator, nitrate anion. The more pronounced trends occurred in both sodium and potassium solutions with this anion. Different intensities of calcium behavior were also observed depending on the concentration of the nitrate solution. However, they still followed the trend of development in solutions with other anions. 

The decrease in concentration in the nitrate solutions could have been due to the formation of crystalline products during reactions with the anion of these solutions. In the hydroxide and carbonate solutions, after an initial increase of 0.5 mg/L by day 7, the concentration subsequently increased to almost the original concentration by day 14. From day 14 onwards, the concentration decreased by approximately 0.1 mg/L. This development corresponded to a steady-state equilibrium. Therefore, the first 7 days when the calcium concentration increased were very different for the development of nitrate. This may have been due to the dissolution of the portlandite from the beam or the release of the pore solution containing calcium ions. After this increase, the process then decreased and, from day 28 onward, became sigmoidal; equilibrium was assumed. The amount of calcium released into the solution in the potassium nitrate during the measurement was approximately 0.3 mg/L higher than the amount released into the sodium nitrate solutions. The final amount of calcium in the nitrate solution was higher than the original amount by approximately 0.75–0.9 mg/L. This was in contrast to the development of the amount in the hydroxide and carbonate solutions, where the original amount decreased to the final amount by approximately 0.5 mg/L.

The increase in exposure temperature to 40 °C had only a minor effect on the actual behavior of the calcium ion concentration curves in the solutions. The development of calcium in the RC test specimen (TS) was similar at 40 and at 25 °C. The concentration of nitrate in the solutions was also similar in value. At 40 °C, the peak of the increased concentration was 1.4 mg/L; while at 25 °C, the maximum was 1.3 mg/L. The difference in the development of calcium ions in nitrate solutions was the more moderate increase in concentration up to day 7, from which point it increased steeply to the aforementioned peak of 1.3 mg/L. Then concentration decreased but was still higher than the initial concentration. The trend in concentration stabilized at 0.7 mg/L higher than the initial concentration. This applied to 0.5 mol/L nitrate solutions. The behavior of calcium in the 1 mol/L solutions followed the same pattern, only reaching values less than 0.2 mg/L in concentration. The final stabilization of the concentration was found in the diffusion equilibrium of the beam/solution system. The overall behavior of the calcium concentration in nitrate at 40 °C had fewer fluctuations. In addition, the curves of the individual nitrate solutions were less widely spaced. The curves for carbonate and hydroxide concentrations were identical at both temperatures. After an initial decrease by day 7, a subsequent increase by day 14 was observed before a slight decrease and a steadying of the concentration in diffusive equilibrium. The difference in the behavior at higher temperatures was that the initial decrease by day 7 was only half as great at 40 °C (by 0.5 mg/L) as at 25 °C (when the concentration decreased by 0.8 mg/L). Therefore, at 40 °C, with further increases, the concentration rose above the initial concentration, with a peak at 0.85 mg/L after a subsequent decrease, and the concentration then settled at a value 0.3 mg/L higher than the initial concentration in diffusive equilibrium.

The increments in the calcium concentrations in the individual solutions were likely due to the dissolution of products from the hardening of the cement matrix or the leaching of the pore solution from the cement matrix [37]. Concentration losses were more important to this study. They predicted the formation of products containing calcium ions. Such products may be calcite–calcium carbonate (CaCO_3_) formed by air carbonation, and portlandite–calcium hydroxide, Ca(OH)_2_, formed as a byproduct of the hydration process, resulting in the formation of the primary CSH gel [38,39,40]. In addition, CSH or CASH gel may be formed (where A denotes an alkali, this product will be discussed in later sections). It was assumed that these products could partially eliminate diffusion, especially through smaller pores. The formation of portlandite in the form of a crystalline slip on the surface of the beam could then be confirmed by SEM image analysis of the sample [41,42]. There was an equilibrium between these products and the solution, which indicated an overall decrease in concentration from the original. As this decrease in concentration did not reach significant values in the experiment performed (a decrease of 0.2 mg/L), we assumed a small amount of these emerging products. The RC test specimen showed an increase in calcium, likely due to the calcium loss from the beam and likely increased by impurities. In carbonates and hydroxides at 25 °C in RC, this calcium loss to the solution was limited by the formation of the products. Product formation in nitrates was suppressed by the nitrate anion, with which more readily soluble products formed [42]. 

#### 3.2.2. Monitoring the Concentration of Sodium Ions 

The behavior of the sodium concentration in the potassium solution at 25 and 40 °C is shown in Figure 5. For test specimens (TS) at the lower temperature, the concentration of sodium in the ambient solution stagnated or slightly decreased by day 7. This behavior was least noticeable in the potassium nitrate solution, regardless of concentration. In the nitrate solution, this increase reached a maximum of 0.1 mg/L sodium, as compared to the initial concentration. It continued to stabilize at this concentration and did not fluctuate significantly further. The steepest increase with the highest maximum relative to the other samples was achieved by the sodium behavior in 1 mol/L potassium hydroxide. The peak of the initial trend was on day 7 at a concentration 0.7 mg/L higher than the initial concentration. However, this increase decreased by approximately 0.6 mg/L by day 14 when sodium was consumed in the solution. The concentration stabilized at approximately 0.1 mg/L. The behavior of the concentration in the 0.5 mol/L hydroxide solution was different. The initial increase in concentration was only 0.3 mg/L on day 7 and the concentration then decreased by 0.15 mg/L. However, from a concentration 0.15 mg/L higher than the initial concentration, it then increased again to 0.3 mg/L on day 84. Then it decreased until the end of the measurement. Product formation and the possible dissolution could then be assumed. The trend of the calcium concentration in the carbonate solution followed the same trend as in 0.5 M potassium hydroxide. The potassium carbonate solutions showed an increase in sodium concentration up to day 14 when, in the 1 mol/L solution, the increase between days 7 and 14 was gradual and reached a maximum increase of 0.5 mg/L. The concentration in the 0.5 mol/L carbonate solution was 0.1 mg/L lower throughout the development. From day 14, the concentration decreased until day 56. From this day onwards, there was an increase in the concentration to a value corresponding to the original maximum of 0.5 and 0.4 mg/L. Thereafter, the concentration decreased. At 40 °C, the overall pattern of the individual curves changed. The most significant changes were observed at the very beginning of the exposure to each medium until approximately day 28. Samples of K_2_CO_3_ 1 mol/L and KOH 0.5 mol/L showed a completely different pattern of behavior, but repeated measurements have eliminated any error in preparation or determination by obtaining identical results. The remaining environments showed a similar character, and the behavior of the sodium content in the system of nitrate or hydroxide solution and TS at 40 °C was very simple and described a simple diffusion with an equilibration of concentrations. After immersion of the beam in the solution, there was a slight fluctuation of the sodium concentration into positive and negative values up to day 28. This fluctuation was attributed to the establishment of equilibrium. On day 28, the concentration stabilized at 0.1–0.2 mg/L higher than the initial concentration. This corresponded to the sodium concentration from the material sheet of the used cement.

While monitoring sodium ions in sodium solutions, very pronounced curves were observed in the first half. The most pronounced changes in the concentrations over the experimental period were observed for the sodium carbonate solution with concentrations of 0.5 and 1 mol/L at 25 °C. This behavior was attributed to the gradual formation and transformation of secondary products on the TS surface, which appeared to be readily soluble. The concentration of 1 mol/L sodium hydroxide solution decreased significantly from the initial 23 to 11 mg/L by day 7 of observation and then decreased to the final 8.5 mg/L; the diluted solution, 0.5 mol/L, behaved similarly with approximately half of the concentration changes. Significant decreases in concentration were observed in the nitrate solutions, where the concentration of sodium ions in the 0.5 mol/L nitrate solution steadily decreased from approximately 23 to 3 mg/L from day 1 to day 28 of the experiment, respectively, followed by a further decrease likely due to the formation and dissolution of one of the crystalline products on the TS surface. The 1 mol/L nitrate solution then decreased steadily to final values close to zero confinement. The apparent depletion of sodium ions in this environment was then demonstrated in the following sections by the strong coverage of TS by a layer of crystallized nitrate.

In the simultaneous experiment conducted at an increased temperature of 40 °C, we observed a more pronounced fluctuation in the curves of the carbonate solutions. The progression of the individual nitrate curves was similar at both temperatures studied, including the values of the ion concentrations achieved in the respective solutions, with one exception, namely the NaOH solution of 1 mol/L concentration. A much higher loss of the concentration of this ion was noted. During deposition at a higher temperature, a gel-like product was formed on the surface of TS. 

#### 3.2.3. Monitoring the Concentration of Potassium Ions 

Potassium is quite closely related to sodium, as indicated by its position on the periodic table, its reactivity, and many other properties. As partially outlined by the results in the previous section, similar behavioral dependencies were observed when monitoring the concentrations of potassium ions as a complementary ion in sodium. The initial assumption that there may be a simple exchange of sodium ions for potassium ions, and vice versa, was completely refuted since the ICP results as well as the SEM-EDX results showed that only a partial replacement occurred and resulted in either the formation of double crystalline products in the environment or the formation of a C(NaK)SH hydration gel. 

For TS road cement, the behavior of potassium at 25 °C in sodium solutions was not readily describable, as shown by the time course of concentration changes in Figure 6 left. In all solutions, the potassium concentration gradually increased until day 14 or 28, when it reached a concentration of approximately 0.4 mg/L. From day 28 onwards, the potassium concentration in the different solutions evolved with different intensities but still at positive values as compared to the initial concentration. The potassium concentration in 1 mol/L sodium carbonate changed most intensively, reaching values 0.83 mg/L higher than the initial value. The least intense development was in 0.5 mol/L sodium hydroxide, where the concentration increased by 0.27 mg/L. The other concentrations were between the maximum concentrations of the two solutions. The behavior of the potassium content of all solutions corresponded to the establishment of equilibrium.

The potassium content of both solutions finished at concentrations lower than the initial values. The concentration was stabilized at values up to 15 mg/L lower. At 25 °C, the trend in all hydroxide and nitrate solutions with the RC beam was a sinusoid shape fluctuating around the original concentration. The trend was then most pronounced in the 1 mol/L solutions, but the trend was also followed by the 0.5 mol/L solutions. This suggested the establishment of the concentration in the system, and there were no significant concentration trends suggestive of product formation. At 40 °C, by day 56, the potassium evolution in all nitrate and hydroxide solutions was below the original concentration. At both temperatures, the evolution of the curves describing the potassium ions in potassium solutions had a similar character to that of the sodium ions in the previous section. That is, the curves of the carbonates reached significant fluctuations in the growth and decline of the potassium ion concentrations in the respective media due to the formation and subsequent dissolution of crystalline products on the TS surfaces. The action of hydroxides and nitrates then showed only minor variations. The 0.5 mol/L solutions, after an initial decrease of 6.5 mg/L, remained at this concentration with fluctuations. In the curves shown in Figure 6 right, TS deposition in solutions at 40 °C showed changes in the concentration of this element with opposite trends in terms of higher concentration changes at the initial days of exposure and, conversely, lower maximums in the last stages of this experiment. The behavior of the potassium content in nitrate was accompanied by an initial increase during the first seven days. By the seventh day, this increase reached a concentration of 0.75 mg/L in 1 mol/L solution. The concentration in the 0.5 mol/L solution was approximately 0.1 mg/L lower at this maximum. On day 28, the concentration decreased to below the initial value of approximately 0.2 mg/L in the 1 mol/L solution. After a slight increase, when the concentration of potassium in the 0.5 mol/L solution was equal to the initial concentration, the concentration decreased further by day 120, as compared to the initial value. The potassium concentration in the 0.5 mol/L solution was 0.1 mg/L different from that in the 1 mol/L solution. The concentration of potassium in the carbonates and hydroxides varied from the original concentration by a maximum of ±0.1 mg/L. The highest increase was in sodium carbonate at 1 mol/L. The lowest concentration was reached in 0.5 mol/L sodium hydroxide. Potassium concentrations in all three solutions clashed at approximately 0.6 mg/L lower than the original concentration on day 56. Further, in the hydroxides, the potassium concentration remained lower than the original, and there was no apparent difference in the behavior of the concentrations in the differently concentrated solutions. In the carbonate solutions, the potassium concentration increased slightly from day 56 to day 120. The potassium concentration in 1 mol/L carbonate solution was approximately 0.6 mg/L higher than that in the 0.5 mol/L solution.

For the evaluation of the sodium ion evolution, if a catex effect occurred, there would have to be sodium evolution and potassium loss in the same solution (or vice versa). This process was evident in some nitrate and hydroxide solutions at higher temperatures. According to the curves, most of the processes corresponded only to the steady-state concentration of the system. This common fluctuation was attributed to the formation of gels or unstable salts. The carbonate curves then, as in the previous cases, had a high intensity of evolution, likely due to carbonation. This was also shown by the contradictory development of calcium. At different temperatures as well as at different concentrations, the trend curves were the same, but they differed only in the intensity of the individual dips and increases in concentration. All decreases in potassium concentrations were then suggested by the dropout of salt crystals on the TS surface, which was confirmed by the SEM images. Since there was a common trend with the evolution of sodium ions in solution, the formation of double salts was assumed. However, it should be mentioned that the gel formation (CASH) would be accompanied by its subsequent expansion by water adsorption. This expansion would also lead to the formation of pores and the possible formation of blooms on the surface (which could not be expected in such a short time interval as the experiment).

### 3.3. Monitoring the Depth of Diffusion of Alkali Ions into the TS Structure Using EDX Mapping

This section describes the evolution of the permeation of alkali ions into and further through the cement matrix. Although the monitoring of the changes in the concentration of the alkali solution using the ICP-OES method yielded very sensitive and representative results, the changes in the concentration of the respective alkali in a given solution may not be definitely or obviously related only to the permeation of the alkali into the TS but may form crystalline and amorphous products on the surface of the TS or on the walls of the hermetically sealed vessels used.

From the EDX mapping results, two basic trends of observations emerged. The first one, common to all the studied samples and solutions, was that, with increasing concentrations of the alkaline solution in which the TS was immersed, as well as with increasing temperature of the experimental conditions, a more significant alkali invasion inside the TS occurred. This result was quite expected and also consistent with the results of previous studies on the permeation of chlorides and sulfates as degradation media [43,44,45,46,47,48,49]. The second result of the observation was the ability of sodium ions to penetrate deeper into the sample mass than was the case with the potassium analog. This also underlined the theoretical assumption of ion mobility and the associated ability to migrate, as described previously in studies on pure aqueous solutions [50,51]—papers or in [52,53]—book chapters.

The overall view of permeating alkali ions into the structure of cement composites over time was captured by the EDX mapping images. Based on the different types of alkali anion solutions in which the TS had been deposited, quite large differences were observed.

In carbonate solutions, alkali permeation occurred only a few millimeters below the sample surface, which was due to a combination of several different factors such as surface carbonate degradation, partial CASH gel formation below the surface, or crystallization of the reagent in the pores and micropores of the TS. This carbonate degradation induced the formation of calcite from portlandite and a general decrease in alkalinity in the affected area, as also described in [54,55,56]. In the images of the TS microstructure in Figure 7, all these factors were observed.

The action of alkaline nitrate solutions represented great potential for the description of alkali diffusion. Even the results from the ICP method itself, described in the sections above, suggested significant amounts of migrating alkali. According to EDX mapping, there was a significant amount of alkali diffusion inside the TS, but no significant changes in structure were observed that suggested the possibility of CASH gel formation, as captured by the detailed microstructure images in Figure 8. Only the crystallization of the nitrate in the form of fine needles or flat formations in combination with portlandite was evident. The composition of these products was analyzed by EDX and only showed the corresponding nitrate or double salt with calcium. 

Hydroxides appeared to be the most aggressive environment for alkali penetration into the TS structure according to the results obtained. In particular, the combination of an elevated temperature of 40 °C and an increased concentration of 1 mol/L resulted in a very rapid, widespread penetration that passed through the entire sample over time. However, not only the penetration into the sample occurred but also the action of the sodium or potassium hydroxide solution induced quite significant changes to the structure of the samples themselves and a gradual transformation of the CSH gel into a CASH gel of very fine structure. This resulted in a reduction in porosity in the first stage, and a CASH gel was also formed, which then created high stress within the samples and thus gradually broke down the mechanical properties. A snapshot of the massively transformed internal CSH gel structure is illustrated in Figure 9.

The graph in Figure 10 shows the reduction in porosity due to TS deposition in all the ionic solutions used, namely nitrate, carbonate, and hydroxide. The trend of the above graph showed that the most significant reduction in porosity was achieved by the effect of hydroxide, followed by nitrate, and then carbonate. Overall, there was a reduction in TS porosity of up to 1.2%.

The images in Figure 11 show the gradual changes in the concentration of sodium ions in the TS. In TS subjected to a carbonate solution, the gradual formation of a sodium-enriched shell due to the carbonate degradation phenomenon described above was observed. TS exposed in nitrate solution showed a very rapid saturation of the entire sample volume with sodium ion and showed little change over time. The most pronounced sodium permeation of TS resulted due to the overexposure to a sodium hydroxide solution, where a gradual permeation of TS alkali was evident from the images. At the same time, the presence of a heavily sodium-permeated region was evident where the CASH gel shown in Figure 9 was present.

### 3.4. Formation of Secondary Products in Solutions and on TS Surfaces

As already outlined in the discussion of the results in the previous two sections, during the deposition of TS in solutions with alkaline ions, there were observable changes on the surfaces of the observed TS. With increased TS exposure in the alkaline ion solutions used, secondary product formation occurred at varying degrees of intensity. The degree of intensity was directly dependent on the concentration of the alkali solution used as well as the temperature of the environment in which these samples were stored. The following discussion and analyses were based on the results of the observations and the determinations of these products for an environment with an initial concentration of 1 mol/L of the alkali in question and a temperature of 40 °C for the greatest clarity; however, control determinations were made on all samples.

Magnified images of the products on the TS surfaces were captured and are shown in Figure 12. This was described by subsequent analyses, in particular by SEM-EDX, as a sodium-silica gel covering the entire surface of the TS. Another minor difference in the migration of ions between the solution and TS at different temperatures was the longer-term trend of equilibrium settling. In the environments of sodium and potassium carbonate, during the exposure of TS, the gradual formation of a “skin” and the coverage of the surface with a fine product consisting, according to analysis, of an amorphous phase (approximately 55%) and then of the crystalline carbonate form of the respective solution in which the TS under study was exposed, as well as of crystalline carbonates, either sodium carbonate or calcite. The fine structures of these surfaces were then observed in the SEM images. Figure 13A represents the structure of the amorphous phase of the formed gel with very fine particles of crystallized carbonates, while Figure 13B shows the crystals separately.

On the surface of samples deposited in both sodium and potassium hydroxide solutions, a gradual coalescence of an amorphous, gel-like phase was observed, with an elemental composition, as well as morphology, that demonstrated the formation of a probable secondary C(A)SH gel. Sporadically, plate-like crystals of portlandite occurred in the solution or on the TS surface. Over the course of the experiment, an increasingly significant, very fine, porous-to-spongy phase of amorphous CASH gel on the TS surfaces was observed in the SEM image in Figure 13C, and sporadically occurring, large flat crystals of portlandite are depicted in the image on the right in Figure 13D.

The products formed on TS surfaces in an alkali nitrate environment could generally be classified as crystalline, sharp-edged, as shown in Figure 13E. Very fine hair-like to needle-like crystals were also found and were identified by semi-quantitative XRD and EDX analyses as the crystallized salt of the respective medium and as sodium and potassium nitrate, sodium and calcium doublets, as shown in Figure 13F. No form of alkali-silica gel was present on the surface of these TS.

These inclusions of the TS surface led both to a blockage of the surface itself but also to the inclusion of pores up to a depth of several (even tens) of micrometers, thus preceding the penetration of the medium to a greater depth of the TS structure. For this reason, changes were observed on the curves showing the ion concentration fluctuation across even longer time periods, but the total alkali content of the TS did not change further, as demonstrated by the evaluation of the EDX mapping.

## 4. Conclusions

In this work, the phenomenon of permeation of alkali ions, namely sodium and potassium, into cement composites prepared from road cement was studied. Other experimental modes studied were variables such as increased exposure temperature, increased solution concentrations, and the type of anions in the solution.

Both sodium and potassium ions in all the three environments studied, namely carbonate, hydroxide, and nitrate, penetrated into the composite and further into its structure by different mechanisms.The driving force was the increased temperature of the experiment as well as the increased concentration of the solution to which the TS was exposed.In sodium or potassium nitrate exposure, the penetration of the alkaline ion into the TS and the permeation of the entire sample volume occurred most efficiently, and this occurred more rapidly with sodium nitrate due to sodium ions having a higher mobility than potassium ions.The action of carbonates and hydroxides transformed the original and traditional hydration products into a modified alkaline-silica gel–CASH gel, while for carbonate, this phenomenon was minor. The most significant consequence of the action of carbonate solutions was the formation of a shell close to the surface as a result of carbonate degradation associated with calcite formation, which reduced the alkalinity of the surroundings to the neutral region, which was thought to prevent the formation of ASR. On the contrary, this occurred when hydroxides were present, when this highly alkaline environment was gradually permeated throughout the entire volume of the TS and a very intense transformation of the original CSH gel into a modified CASH gel or even an ASR gel occurred.

## Figures and Tables

**Figure 1 materials-15-04730-f001:**
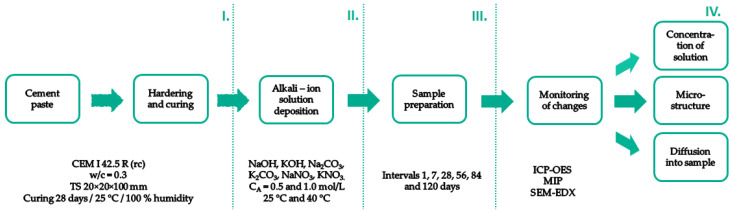
Schematic workflow of the experiment.

**Figure 2 materials-15-04730-f002:**
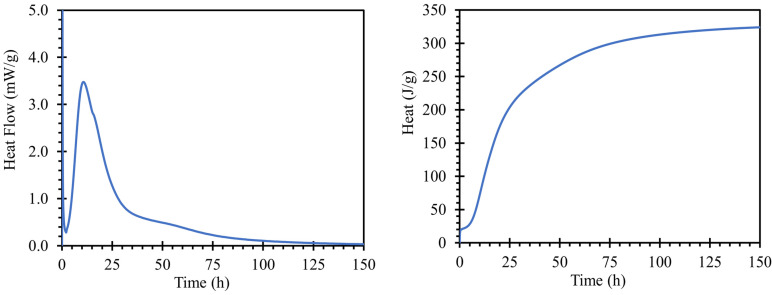
Hydration kinetics of used road cement using water to cement ratio of 0.3 recorded by means of isothermal calorimetry.

**Figure 3 materials-15-04730-f003:**
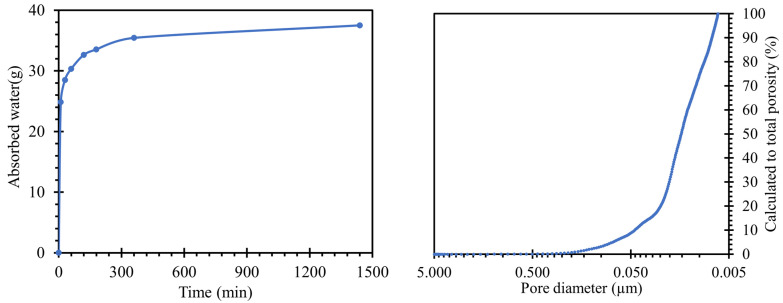
**Left**—results of the water absorption test of the test specimens indicating the amount of water absorbed over time; **right**—determined porosity, converted to total porosity in the range of 0.005 to 5 µm, measured by MIP.

**Figure 4 materials-15-04730-f004:**
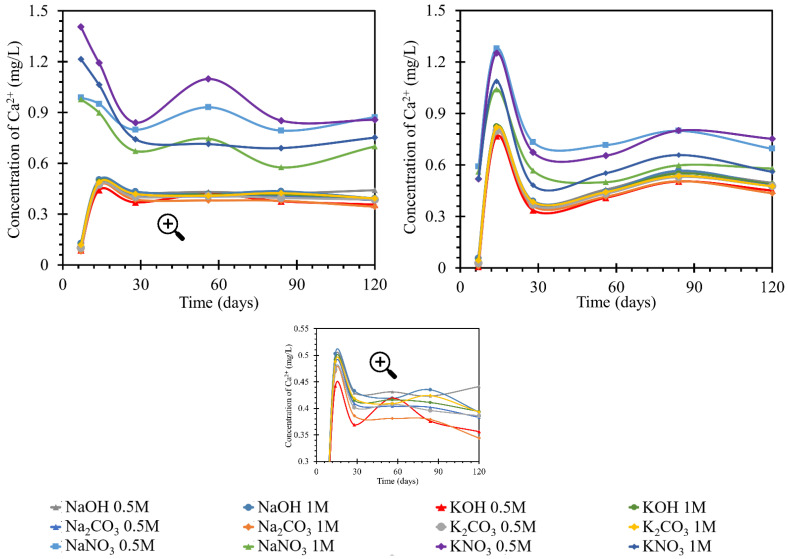
Results of monitoring calcium ion concentrations in individual solutions, on the left at the laboratory temperature of 25 °C, on the right at 40 °C. The Ca^2+^ concentrations were determined using the ICP-OES method; for clarity, the error bars are not shown, but they reach a 1.83% maximum.

**Figure 5 materials-15-04730-f005:**
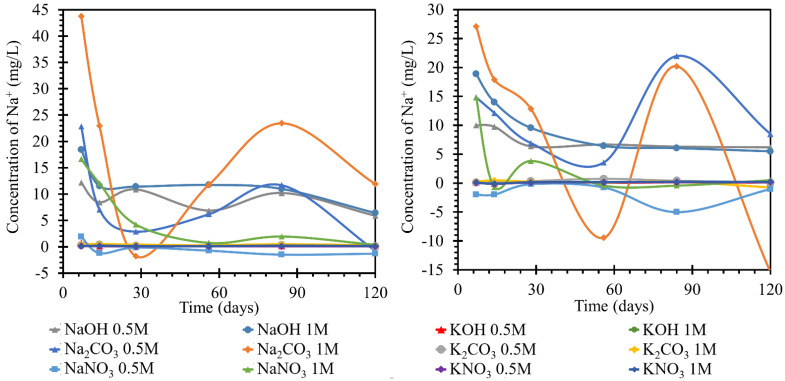
Results of monitoring sodium ion concentrations in individual solutions, on the left at the laboratory temperature of 25 °C, on the right at 40 °C. The Na^+^ concentrations were determined using the ICP−OES method; for clarity, the error bars are not shown, but they reach 1.91% maximum.

**Figure 6 materials-15-04730-f006:**
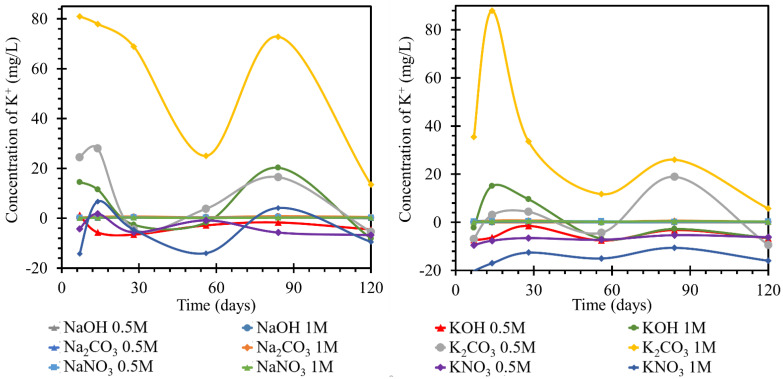
Results of monitoring potassium ion concentrations in individual solutions, on the left at the laboratory temperature of 25 °C, on the right at 40 °C. The K^+^ concentrations were determined using the ICP−OES method; for clarity, the error bars are not shown, but they reach 1.85% maximum.

**Figure 7 materials-15-04730-f007:**
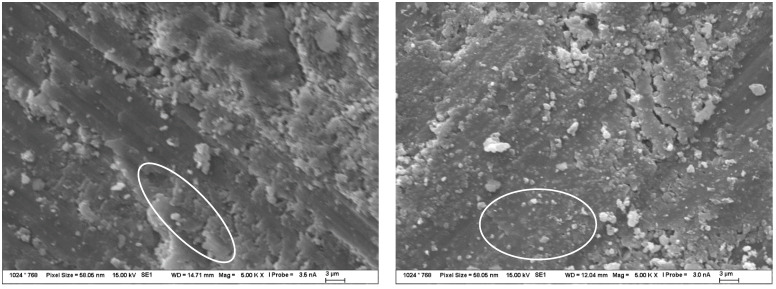
Structure of TS after 120 days of exposure in 1 mol/L Na_2_CO_3_ solution at 40 °C—observed by SEM. The image on the left shows CASH gel formation and crystallized calcite to a small extent, while the image on the right shows mainly crystallized carbonate solution and calcite particles formed. The composition of these products was determined by EDX.

**Figure 8 materials-15-04730-f008:**
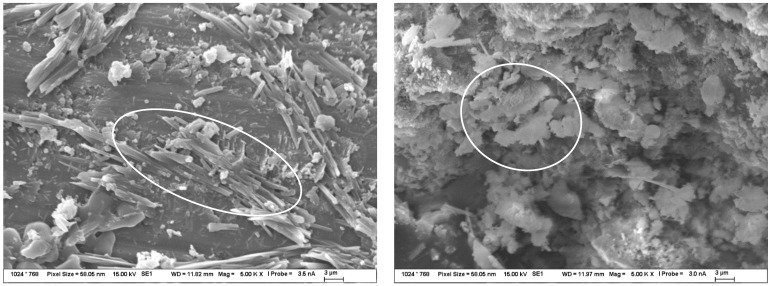
Structure of TS after 120 days of exposure in 1 mol/L NaNO_3_ solution at 40 °C—observed by SEM. Both pictures show crystallized nitrate needles (highlighted by white ellipses), the picture on the right shows the structure of the double salt with calcium formed by the reaction on the original portlandite plates.

**Figure 9 materials-15-04730-f009:**
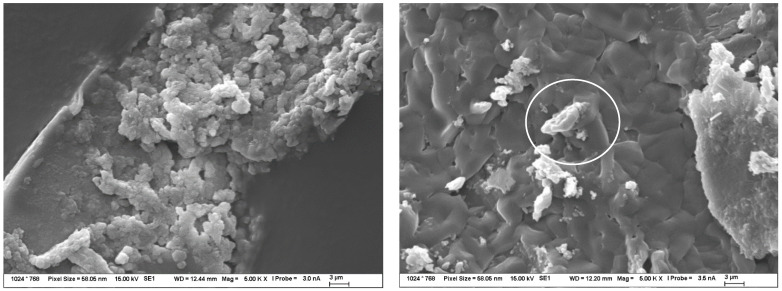
Structure of TS after 120 days of exposure in 1 mol/L NaOH solution at 40 °C, observed by SEM. The images show the very fine structure of the new CASH gel without crystalline products (marked by white circle).

**Figure 10 materials-15-04730-f010:**
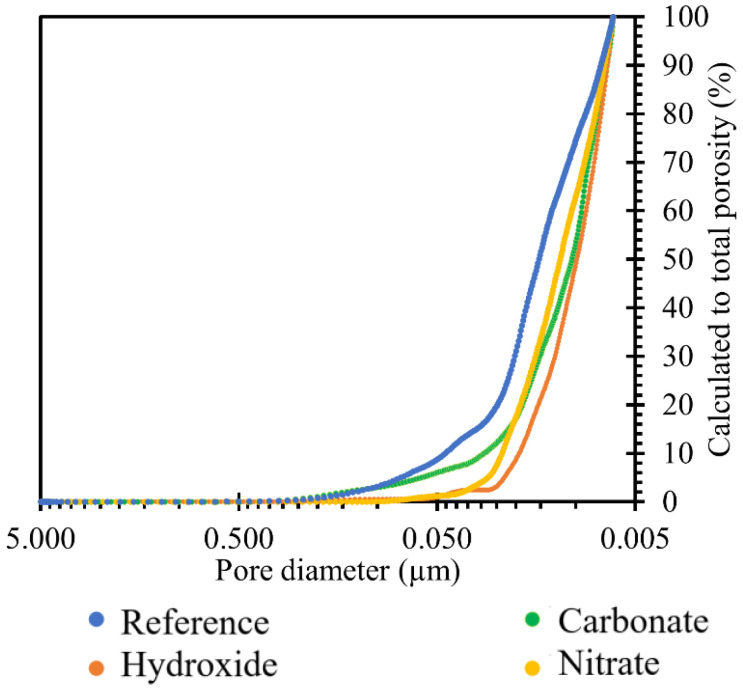
Results of comparison of TS porosity after 120 days of exposure in the respective sodium environment at 40 °C.

**Figure 11 materials-15-04730-f011:**
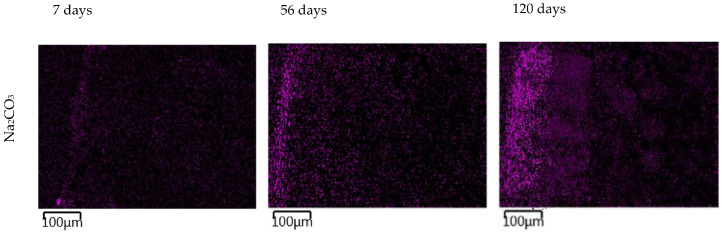
EDX mapping of sodium ion permeation in a 1 mol/L solution at 40 °C over time. The magenta color represents sodium in the EDX mapping images.

**Figure 12 materials-15-04730-f012:**
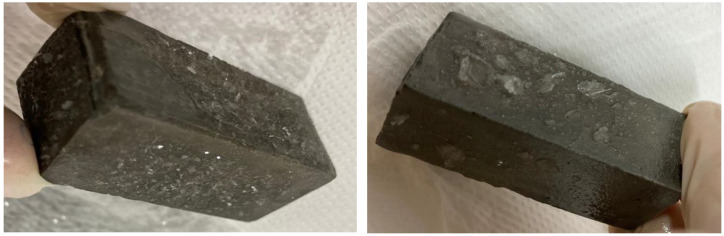
Photographs of products formed on TS surfaces after deposition for 120 days at 40 °C in 1 mol/L solutions, on the left TS deposited in nitrate solution, on the right in hydroxide.

**Figure 13 materials-15-04730-f013:**
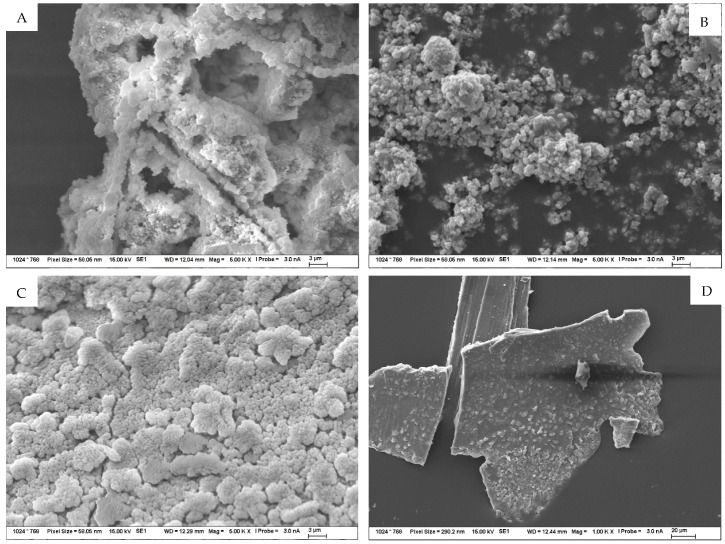
(**A**–**F**) Detailed SEM images of the products formed on the TS surface as a result of exposure to 1 mol/L solutions at 40 °C for 120 days. Figure (**A**,**B**) symbolize structure of TS exposed to carbonate solution; (**C**,**D**) exposed to hydroxides and (**E**,**F**) exposure to nitrates.

**Table 1 materials-15-04730-t001:** Chemical composition of the cement according to EN 196-2. The abbreviations UR and LOI stand for unsoluble rest and loss on ignition, respectively.

CaO	SiO_2_	Al_2_O_3_	Fe_2_O_3_	MgO	SO_3_	Cl^–^	K_2_O	Na_2_O	Na_2_O ekv	UR	LOI
63.32	20.33	4.57	3.52	1.52	3.18	0.028	0.73	0.15	0.63	0.41	1.61

**Table 2 materials-15-04730-t002:** Phase composition in wt.% of used cement. Determined by the mean XRD. The meaning of the abbreviations used: C_3_S—tricalcium silicate, C_2_S—dicalcium silicate, C_3_A tricalcium aluminate, C_4_AF—tetracalcium aluminoferrite, CaO—lime, CaCO_3_—calcite, and CŜH—gypsum.

C_3_S	C_2_S	C_3_A	C_4_AF	CaO	CaCO_3_	CŜH	Amorphous
69.25	10.89	6.52	10.96	NA	NA	2.38	NA

## Data Availability

All the data is available within the manuscript.

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
