# Peer review of "Monitoring of Ion Mobility in the Cement Matrix to Establish Sensitivity to the ASR Caused by External Sources"

_materials, 2022, doi:10.3390/ma15144730_

Round 1
Reviewer 1 Report
- The literature survey of the paper in section 1 needs a substantial further refinement in the aspect of the correlation of ion migration into the cementitious matrix with the risk of alkali silica gel formation. In the present form the scope of literature survey is limited to establish a clear research gap to be addressed in the present work.
- Line 87-89. The term “to monitor mobility of selected ions” of the objective statement is vague in meaning and need to be fine tuned to provide an objective which is more precise.
- The Author is advised to provide a further explanation on how the monitoring of ion mobility from external sources into a cementitious matrix to predict possibility of ASR formation can contribute to solve the ASR problem of cementitious matrix.
- In section 3.1, line 154, what does the author means by “objected cement paste”?
- In section 3, there were detailed elaboration on the data of the monitoring of alkaline ions exchange between the environment and the specimen, monitoring of the depth of alkaline ions diffusion into the TS structure via EDX mapping, and the formation of secondary hydration products in solutions on the TS surface. The correlation of alkaline ions exchange between the environment and the specimen and the tendency of the ASR degradation is not clearly presented. Therefore, the author is advised to provide a dedicated discussion on The correlation of alkaline ions exchange between the environment and the specimen and the tendency of the ASR degradation of cementitious composite examined.
- The conclusions presented in section 4 can be condensed further by focusing on how the outcome of the work enable prediction of the possibility of ASR degradation through monitoring the mobility of certain ions from the exposure environment into the cement matrix.
Author Response
Dear reviewer,
Thank you very much for your precious and valuable comments on the manuscript. As a junior researcher, these insights were enriching and of great help to me.
A comprehensive language proofreading has been ordered by the journal editors and is not yet incorporated into this manuscript; however, I still hope that the factual relevance is described enough for the review, and the language proofreading should take place in the next 7-10 days.
I have implemented your comments in the manuscript as follows:
The literature search was supplemented by the risks of ASR formation, some options to counteract the influence of ASR and previous research on the mobility of alkali ions through the structure of cement composites.
The term "objected cement paste" is meant the studied composition of the prepared cementitious composite, the test specimen.
In Section 3, dealing with EDX analysis, all the products discussed are identified on the basis of elemental composition and stoichiometry; this is not simply an image-based evaluation. All results are supported by an adequate number of spatterings from different TS sites so that the statements in the text correlate completely with the measured data.
Should I include all these analysis results (these are more than 50 spectra) in the appendices or? I can't think of a suitable reasonable format to insert these EDX analysis results into the paper, so as not to reduce the clarity of the text and results.
The conclusion of the submitted manuscript has been simplified and reworked into individual points so that the individual sub-conclusions are more obvious and easier to understand.
Děkuji mnohokrát za velmi užitečné rady a čas, který věnujete kontrole. Věřím, že všechny druhy těchto rad a postřehů povedou ke zlepšení interpretace a kvality dalších zaslaných rukopisů.
Přeji vám dobré dny, hodně zdraví, radosti a úspěchů.
S pozdravem
Michal Marko

Reviewer 2 Report
Dear authors, please find attached my comments

Author Response
Dear reviewer,
Thank you very much for your precious and valuable comments on the manuscript. As a junior researcher, these insights were enriching and of great help to me.
A comprehensive language proofreading has been ordered by the journal editors and is not yet incorporated into this manuscript; however, I still hope that the factual relevance is described enough for the review, and the language proofreading should take place in the next 7-10 days.
I have implemented your comments in the manuscript as follows:
The literature search was supplemented by the risks of ASR formation, some options to counteract the influence of ASR and previous research on the mobility of alkali ions through the structure of cement composites.
In Section 3, dealing with EDX analysis, all the products discussed are identified on the basis of elemental composition and stoichiometry; this is not simply an image-based evaluation. All results are supported by an adequate number of spatterings from different TS sites so that the statements in the text correlate completely with the measured data.
Should I include all these analysis results (these are more than 50 spectra) in the appendices or? I can't think of a suitable reasonable format to insert these EDX analysis results into the paper, so as not to reduce the clarity of the text and results.
Thank you so much for your very helpful advice and the time you spend reviewing. I believe that all kinds of this advice and insights will lead to improvements in the interpretation and quality of other submitted manuscripts.
Wishing you good days, good health, joy and success.
Best regards,
Michal Marko

Reviewer 3 Report
I have read the Manuscript entitled " Monitoring of Ion Mobility in the Cement Matrix to Estabilish Sensitivity to the ASR Caused by External Sources" and found it very interesting.
In my opinion the work is comprehensive and attractive, however some minor modifications are required. My specific comments are as follows:
- In the introduction, some contents related to the harm of ASR can be added, such as the crack formation. Also, previous studies on reducing ASR can also be discussed in Introduction. Here are some articles for your reference, including “Influence of reactivity and dosage of MgO expansive agent on shrinkage and crack resistance of face slab concrete (https://doi.org/10.1016/j.cemconcomp.2021.104333), “Alkali silica reaction: A view from the nanoscale (https://doi.org/10.1016/j.cemconres.2021.106652)”, and “Alkali-silica reaction (ASR) in the alkali-activated cement (AAC) system: A state-of-the-art review (https://doi.org/10.1016/j.conbuildmat.2020.119105)” .
- There are some grammatical mistakes in some sentences, such as lines 25-27. Please carefully check the whole article.
- Porous characteristics of cement matrix are introduced from line 53 to line 60. To explain it in a more detailed and specific way, the author can refer to the related works to improve it, such as “High-ferrite Portland cement with slag: Hydration, microstructure, and resistance to sulfate attack at elevated temperature (https://doi.org/10.1016/j.cemconcomp.2022.104560)”, “Structure, fractality, mechanics and durability of calcium silicate hydrates (https://doi.org/10.3390/fractalfract5020047)”, “The influence of a novel hydrophobic agent on the internal defect and multi-scale pore structure of concrete (https://doi.org/10.3390/ma14030609)”, and “Fractal analysis on pore structure and hydration of magnesium oxysulfate cements by first principle, thermodynamic and microstructure-based methods (https://doi.org/10.3390/fractalfract5040164)” .
- The significance of the study should be added at the end of introduction.
- The notation of CSH are not consistent in the manuscript. Please check and make corrections.
- In section of Material and methods, the author can add a schematic figure of the experimental setup to make it easier to be understood.
- The abscissa notation of Figure 1 are missing.
- In figure 3, some curves are close to each other. The author can make snapshot of these curves to make them distinct from each other.
Author Response
Dear reviewer,
Thank you very much for your precious and valuable comments on the manuscript. As a junior researcher, these insights were enriching and of great help to me.
A comprehensive language proofreading has been ordered by the journal editors and is not yet incorporated into this manuscript; however, I still hope that the factual relevance is described enough for the review, and the language proofreading should take place in the next 7-10 days.
I have implemented your comments in the manuscript as follows:
The literature search was supplemented by the risks of ASR formation, some options to counteract the influence of ASR and previous research on the mobility of alkali ions through the structure of cement composites.
Minor errors pointed out by you have also been corrected and a simplified graphical diagram of the experiment has been added.
Thank you so much for your very helpful advice and the time you spend reviewing. I believe that all kinds of this advice and insights will lead to improvements in the interpretation and quality of other submitted manuscripts.
Wishing you good days, good health, joy and success.
Best regards,
Michal Marko

Reviewer 4 Report
The manuscript entitled "Monitoring of Ion Mobility in the Cement Matrix to Estabilish Sensitivity to the ASR Caused by External Sources" presents an interesting experimental study conducted on concrete durability in alkali solutions. However, the introduction section includes general affirmation without a quantitative evaluation of previous literature, and other issues must be addressed. The paper needs minor revisions before it is processed further, some comments follow:
The abstract is written qualitatively. The majority of the qualitative statements should be modified for quantified result comparisons. Currently, the abstracts present only the materials and methods.
The introduction section must be improved. The are multiple general affirmations in the first paragraph that are not related to the literature (there are no citations to support this affirmation). Therefore, please introduce corresponding citations, or move the paragraph to the end of the section (it could be presented as a conclusion of the presented state-of-the-art).
". [1–3]" – The citations should be positioned before "." Please make corresponding corrections in the entire manuscript.
"During clinker firing, sulfates most often combine with alkalis." – cite corresponding studies.
In the introduction section, a comprehensive and exhaustive review of the state of the art in the field of the study must be provided. Please refer to previous works, and highlight the experiments and results published previously. In the current form, the introduction section only provides basic/general information about the chemical and structural properties of cement. Please provide a short description of previous studies that evaluate these properties.
Please avoid group citations in one phrase, such as [1-3], [1,2,4], etc. Please discuss the highlights individually.
Please evaluate suitable literature and clearly state their contributions, pros and cons and how the current work would address them. Lines 44-67 include multiple affirmations without a clear background in the literature. Please introduce citations at the corresponding position to assure a clear correlation between the affirmations and the supporting study.
Table 1 - two types of iron oxides have been detected in this type of material, therefore, please replace Fe2O3 with FexOy or provide the scientific proof to support your results.
Table 1 – The sum is different than 100, please make corresponding adjustments.
Figure 1- divide in two figure captions: Figure 1, a) and Figure 1, b), not left and right.
Line 162 – "EN 196-", replace with "EN 196-3:2016"
Figures 7 to 12 – please introduce figure labels to highlight the areas of interest for the readers.
Conclusion:
Please improve the conclusions and present them following the main recommendations by Academia of giving the conclusions of the study by points with highlights.
English grammar and spelling check mandatory:
Estabilish, replace with Establish in the title,
Table 1. Chemical composition in of the cement – remove "in"
Replace "," with "." were necessary Etc.
Author Response
Dear reviewer,
Thank you very much for your precious and valuable comments on the manuscript. As a junior researcher, these insights were enriching and of great help to me.
A comprehensive language proofreading has been ordered by the journal editors and is not yet incorporated into this manuscript; however, I still hope that the factual relevance is described enough for the review, and the language proofreading should take place in the next 7-10 days.
I have implemented your comments in the manuscript as follows:
The literature search was supplemented by the risks of ASR formation, some options to counteract the influence of ASR and previous research on the mobility of alkali ions through the structure of cement composites; the necessary literature has also been mined so that even general statements are sufficiently scientifically supported
Minor errors pointed out by you have also been corrected and a simplified graphical diagram of the experiment has been added.
Table 1 shows the chemical composition of the cement used, according to the standards and in comparison with the work of other authors, the total amount of iron was expressed as Fe2O3, but it is obvious that this material contains many other iron oxides. Substituting Fe2O3 for FeyOz would be preferable from the point of view of complexity, but under this assumption the exact content cannot be given as the exact stoichiometry is not known. The error in this table where the sum is not equal to 100 has been removed (it was due to rounding).
The labeling of the figures "left and right" was recommended to me in a previous review of another publication, but if you insist on the "a and b" labeling, it is a mere trifle for me to revise the labeling.
Thank you so much for your very helpful advice and the time you spend reviewing. I believe that all kinds of this advice and insights will lead to improvements in the interpretation and quality of other submitted manuscripts.
Wishing you good days, good health, joy and success.
Best regards,
Michal Marko

Round 2
Reviewer 2 Report
The authors were adequately responded to the comments. The manuscript can be accepted after editing of English language and text.